# Performance Enhancement of Kaolin/Chitosan Composite-Based Membranes by Cross-Linking with Sodium Tripolyphosphate: Preparation and Characterization

**DOI:** 10.3390/membranes13020229

**Published:** 2023-02-14

**Authors:** S. Bouzid Rekik, S. Gassara, J. Bouaziz, S. Baklouti, A. Deratani

**Affiliations:** 1Institut Européen des Membranes, IEM, UMR-5635, ENSCM, CNRS, Univ Montpellier, 34095 Montpellier, France; 2Laboratory of Advanced Materials, National School of Engineering, University of Sfax, Sfax 3038, Tunisia; 3Bioengineering, Tissues and Neuroplasticity, EA 7377, Faculté de Médecine, Université Paris-Est Créteil, 8 rue du Général Sarrail, 94010 Créteil, France; 4Laboratory of Materials Engineering and Environment, National School of Engineering, University of Sfax, Sfax 3038, Tunisia

**Keywords:** composite ceramic/polymer membrane, phase separation, low-cost and environmentally friendly fabrication, finger-like morphology, pore size adjustment

## Abstract

A new family of environmentally friendly and low-cost membranes based on readily available mineral and polymeric materials has been developed from cast suspensions of kaolin and chitosan using aqueous phase separation and polyethylene glycol as a pore-forming agent. The as-fabricated membranes were further cross-linked with sodium tripolyphosphate (STPP) in order to strengthen the properties of the obtained samples. The functional groups determined by FTIR and EDX confirmed that the reaction occurred. A detailed study of the effects of cross-linking time on the physicochemical, surface and permeation properties showed that a 30-minute reaction enabled the composite membrane to be stable in acidic media (up to pH 2) and increased the mechanical strength twofold compared to the non-cross-linked membrane. A similar morphology to that generally observed in polymeric membranes was obtained, with a sponge-like surface overlaying a finger-like through structure. The top layer and cross-section thicknesses of the membranes increased during STPP post-treatment, while the pore size decreased from 160 to 15 nm. At the same time, the molecular weight cut-off and permeance decreased due to the increase in cross-linking density. These results observed in a series of kaolin/chitosan composite membranes showed that STPP reaction can provide control over the separation capability range, from microfiltration to ultrafiltration.

## 1. Introduction

Membrane technology plays a crucial role in many separation processes, including wastewater treatment and drinking water production. The most commonly used membranes can be classified into two main families based on the material they are made of: ceramic and polymeric membranes. Each of them has its own advantages. Polymeric materials are more versatile in terms of shape and morphology to produce membranes with low-cost processes, which allows them to be tailored to each application, covering the whole range of pore sizes down to dense membranes [1,2]). Ceramic membranes, mainly composed of mineral oxides, have high thermal and chemical stability, making them suitable for a wide range of applications [3]. Their high permeability and long operation lifespan before replacement characterize these membranes thanks to their easy cleaning and stability under harsh conditions. However, the energy-intensive, multi-step manufacturing process, which induces a high investment cost, remains an undeniable limitation to their industrial development, even though the life cycle analysis shows that they could be competitive in many applications [4,5].

Therefore, many efforts are devoted to reducing the overall manufacturing cost of ceramic membranes. A first approach is to replace expensive pure metal oxide powders, such as Al_2_O_3_, ZrO_2_ or TiO_2_, commonly used to prepare ceramic membranes [6] with naturally occurring raw materials, such as clays [7,8,9], phosphates [10], kaolin [11,12] and fly ash [13,14]. Furthermore, it is generally accepted that sintering energy requiring very high temperatures contributes to about 60% of the overall manufacturing cost of ceramic membranes. A recent review describes strategies used to decrease sintering energy by lowering the temperature, process time and number of steps [15].

Ceramic–polymer composite membranes are gaining interest because they can lead to a beneficial trade-off to obtain materials with improved properties by combining some advantages of both types of materials [16]. For example, incorporating ceramic particles into a polymer matrix generally improves the membrane hydrophilicity, resulting in higher permeation flux and fouling resistance [17], and increases its mechanical and thermal stability. It should be noted that the increase in membrane performance requires a high compatibility between the mineral particles and the polymer chains. Thus, a suspension of mineral particles in a polymer solution can lead to membranes with morphologies similar to those of polymer membranes using the non-solvent-induced phase inversion technique. In this promising one-step, low-cost approach, the polymer acts as a binder between the mineral particles, ensuring the mechanical stability of the resulting composite membrane. For instance, a suspension of alumina/polyethersulfone (ca 10/1 wt%) in N-methylpyrrolidone has been subjected to phase inversion using water as the polymer coagulation liquid [18]. Morphologies ranging from a sponge-like structure to a finger-like structure have been obtained depending on the viscosity of the dope solution, adjusted by adding a varying amount of non-solvent. The same morphologies have been observed by other authors using similar preparation conditions in terms of suspension compositions and non-solvent mixtures [19,20].

Environmental constraints and the expected scarcity of fossil fuels make renewable polymers an alternative to conventional petroleum-based polymers. Many polysaccharides have film-forming properties that make them suitable for the preparation of membranes. In this context, we have undertaken a study on kaolin (KO) and chitosan (CS)-based composite membranes [11,21]. KO has been widely studied as a starting material for low-cost ceramic membranes due to its low sintering temperature and availability [15,22]. CS, extracted by alkaline deacetylation of chitin, the second most abundant polysaccharide on Earth, is considered a potential green polymer that can be used in many industrial fields. Indeed, the presence of reactive functions (hydroxyl and amino groups) makes it attractive for physicochemical interactions and subsequent chemical modifications. However, the low mechanical strength, stability and solubility of CS films at acidic pH generally limit their applications to dense membranes in aqueous-organic media [23,24]. Composite materials have been developed to overcome some of the limitations of CS, including in membranes [25], adsorption materials to reduce water pollution [26,27] and biomedical applications [28]. Cross-linking via reactive amino groups with various chemical agents is another strategy to stabilize CS films [29,30,31,32].

A previous study has reported the preparation of novel green KO/CS composite membranes by the phase inversion technique using only water as a solvent and non-solvent. In this case, the driving force of the phase inversion was achieved by the change in pH [11]. The film cast in 0.1 M acetic acid was immersed in 1 M NaOH to induce CS phase separation. The obtained membranes showed the best mechanical properties for a KO:CS mass ratio of 55:45. The surface of the KO particles was negatively charged while the CS chains were positively charged due to protonation of the amine groups (pKa ca 6.5). The resulting ionic cross-linking reaction between CS and KO strongly reinforced the cohesion of the composite material. However, the low permeability observed and the low surface porosity of these composite membranes limit their applicability in filtration. An increase in permeability of about one order of magnitude could be obtained by incorporating in the suspension a small amount of water-soluble polyethylene glycol (PEG) as a pore-forming agent [21]. This formulation resulted in membranes with higher wettability and a very porous finger-like morphology, accounting for the increased permeability.

The objective of this report is to investigate the cross-linking ability of CS in composite membranes to improve their chemical stability, mainly in acidic media. The protonation of amine groups leads to the partial dissolution of CS in water. It has been shown in our previous work that ionic interactions with KO shift the dissolution to a lower pH value, but not enough to allow the use of such membranes in aqueous applications [11,21]. CS can be cross-linked covalently or ionically. Harmful chemicals such as glutaraldehyde and glyoxal are most often used for covalent cross-linking [33]. An alternative is to use genipin, a naturally occurring chemical cross-linker [31]. On the other hand, physical cross-linking agents such as low-molecular-weight polyanions, including citrate, tripolyphosphate (STPP) and sulfate, have also been used to circumvent this problem [27,32,34,35]. STPP was chosen for this study because it is known to possess a high charge density and to be able to diffuse into the CS network during the cross-linking reaction. Thus, the fabrication of a CS/KO/PEG composite cross-linked with STPP seems promising to address applications in membrane technology. Another point to mention is that the cross-linking density leads to a tightening of the pore structure [36,37]. Therefore, this work aims to investigate whether it is possible to control the pore size by altering the cross-linking time and thus obtain a range of CS/KO composite membranes of variable selectivity.

The effect of cross-linking density on the properties of the resulting composite membranes was investigated in terms of surface chemistry, structural morphology and pore size distribution (PSD) using techniques such as water contact angle, zeta potential, ATR-FTIR and microscopic observation. In addition, the thermal, mechanical and chemical stability levels of the prepared membranes were also determined. Finally, filtration properties in terms of pure water permeability and molecular weight cut-off (determined from PEG standards rejection) were presented.

## 2. Materials and Methods

### 2.1. Chemicals

KO (kaolin codex) and CS with a mass average molecular weight (Mw) of 180 kg.mol^−1^ and a degree of acetylation (DA) of 20% were supplied by the Laboratoire des Plantes Médicinales (Tunisia) and France Chitine (France), respectively. Polyethylene glycol (PEG, Mw = 10,000 g.mol^−1^) used as a pore former was purchased from Fluka (Seelze, Germany). The cross-linking agent STPP (analytical-grade) was obtained from Sigma-Aldrich (St. Louis, MO, USA). Acetic acid, hydrochloric acid and sodium hydroxide pellets were laboratory-grade chemicals. PEG with a Mw from 200 to 35,000 g.mol^−1^ (Sigma-Aldrich) and polyethylene oxide (PEO) with a Mw between 100 and 300 kg.mol^−1^ (Fluka) were used for membrane MWCO characterization. Deionized (DI) water (18 MΩ/cm, Millipore Milli-Q) was used to prepare all aqueous solutions.

### 2.2. Composite Membrane Preparation

The pure CS solution and the composite suspensions were prepared following a procedure previously described [11]. Briefly, 4% (*w/v*) of the CS powder was dissolved in 0.1 M acetic acid solution under stirring at a temperature between 30 and 40 °C for 24 h. The desired amount of KO powder was then slowly mixed under stirring to obtain the dope suspensions, named CKx, where x denotes the KO *w/v*%. Finally, 1 *w/v*% of PEG powder was introduced in the CK5 suspension to yield CK5-PEG. The suspensions were degassed for 24 h under vacuum to remove air bubbles and then cast on a glass plate with a 700-micrometer gap casting knife using an automatic coater (K Control coater, Erichsen). The membrane under formation was dried at room temperature to partially evaporate the solvents and then immersed into 1 mol.l^−1^ NaOH. The resulting composite membranes were thoroughly washed in DI water to neutrality and stored in DI water for further use. Cross-linking of CK5-PEG composite membranes was performed by soaking in STPP solution (5 *w/v*%) for 0, 15, 30 and 60 min at 25 °C and pH = 4 (by acidification with HCl) [38]. After the desired reaction time was reached, the modified membranes were thoroughly washed with DI water to remove excess cross-linking agent and stored in DI water. The prepared membranes were named CK5-PEG-STPPn (where n is the cross-linking time in min).

### 2.3. Physicochemical Characterization of Membranes

#### 2.3.1. Membrane Morphology and Pore Size Determination

The surface and cross-section morphology of the obtained membranes was observed by scanning electron microscopy (SEM) (Hitachi S-4500, resolution of 1.5 nm at 15 kV). The samples dried at room temperature were cut into small pieces and coated with a thin layer of Pt by sputtering before SEM analysis. In the case of the cross-section observation, the samples were first immersed in liquid nitrogen before being broken. The mean pore size and pore size distribution were estimated from 5–8 images taken from samples of at least three different membranes by treating the surface SEM images with Gwyddion 2.25 imaging analysis software.

#### 2.3.2. Surface Elemental Composition

A Hitachi scanning electron microscope, model TM 1000, coupled to a chemical microanalysis system by energy-dispersive spectroscopy (EDX) was used to obtain the elemental composition of the membrane surface.

#### 2.3.3. Surface Chemical Analysis

The ATR-FTIR technique was used to study the surface chemical structure of the membranes and the chemical changes induced by the cross-linking reaction. ATR-FTIR spectra (Thermo Fisher Nicolet 710 equipped with a KRS5 Diamond plate) were scanned from 650 to 4000 cm^−1^. A background spectrum was measured before each measurement for subsequent correction, mainly due to carbon dioxide and water vapor absorption.

#### 2.3.4. Mechanical Properties

Dynamic rheological experiments (tensile strength and elongation at break) were performed with an MCR Rheometer (Physica MCR301, Anton Paar, GmbH, Graz, Austria). The temperature was controlled at 25 °C with a CTD180 Peltier system. Pieces of 4 × 1 cm^2^ were cut from different parts of the membranes (at least 3 per formulation). This test was performed on samples in the wet state (membrane stored in water for more than 24 h).

#### 2.3.5. Thermal Stability

Thermogravimetric analysis (TGA) was performed using a Hi-Res TGA 2950 (TA instrument, France) to determine the thermal stability of the prepared membranes. Experiments on about 6 mg were conducted from 25 to 800 °C at a heating rate of 10 °C/min under a nitrogen atmosphere to prevent possible thermo-oxidative degradation.

#### 2.3.6. Chemical Stability in Aqueous Media

The membrane solubility in neutral, alkaline and acidic aqueous media was determined from the weight loss of the dry samples. It was defined by the solubilized dry matter content after 24 h immersion in DI water (pH ca 6) and solutions adjusted to pH 9, 4 and 2. Membrane pieces (20 mm × 20 mm) dried in an oven at 80 °C to a constant weight were immersed in 50 mL of solution. After 24 h contact at room temperature, the membrane pieces were removed and dried to a constant weight in an oven at 80 °C to determine the weight of the unsolubilized dry matter. The measurement of the soluble portion was determined as follows:(1)$$SOL=\frac{Mi-Mf}{Mi}\times 100$$
where *SOL* is the percentage of soluble matter and *M_i_* and *M_f_* are the initial and final sample masses, respectively.

#### 2.3.7. Wettability

Water contact angle (WCA) measurement is the most effective method to follow the change in hydrophilicity and wettability of a membrane surface. Prior to measurements, all membranes were washed with DI water, dried on a paper filter overnight and stored in a desiccator. The WCA value was determined by the sessile drop method. A 10-microliter droplet of DI water was deposited on the membrane surface using a 50-microliter syringe. The contour of the droplet was analyzed using ImageJ analysis software. The contact angle was then determined by interpolation (DropSnake). The reported value is the average of 10 measurements.

#### 2.3.8. Zeta Potential

The membrane surface charge (zeta potential) was determined with the SurPASS electrokinetic analyzer (Anton Paar, GmbH, Graz, Austria) based on the streaming potential method [36]. Membrane samples were mounted in an adjustable gap cell and soaked in 1 mM KCl. The gap was set to approximately 100 μm. The electrolyte solution was circulated in the cell between two membrane pieces. The zeta potential was calculated using the Helmholtz–Smoluchowski equation from the measured flow current as a function of pH (2–10 adjusted by HCl or NaOH). The isoelectric point of the membrane was taken as the pH value at which the membrane surface had a net zero electric charge.

### 2.4. Permeation Characterization of Membranes

Frontal filtration experiments were performed using a stirred dead-end cell (Amicon 8050, Millipore Corporation) with an active membrane surface area of 13.4 cm^2^ following a previously described procedure [36]. The membranes were first conditioned by filtering pure water at 4 bar until a constant flow was observed. Then, the filtration cell was connected to a water tank and pressurized to the desired value using compressed air. The mass of permeate crossing the membrane was recorded using the SartoConnect software at regular time intervals.

#### 2.4.1. Pure Water Permeability

Pure water flux (*PWJ*) was measured by flowing DI water through the membrane system using an applied pressure range of 0.5 to 3.5 bar. The *PWJ* (l/m^2^.h) was calculated using the following formula:(2)$$PWJ=\frac{Q}{\Delta t\times A}$$
where *Q* (L) is the volume of permeate, Δ*t* (h) is the permeation time and *A* (m^2^) is the active membrane area. In each experiment, the permeate flux increased linearly with operating pressure. Pure water permeability (*PWP*) was determined from the slope of the linear variation of *PWJ* with applied pressure.

#### 2.4.2. Molecular Weight Cut-Off (MWCO) Determination

The MWCO is defined as the molecular weight of a neutral solute with 90% membrane rejection. Retention experiments were performed using 1 g/l aqueous solutions of model solutes (PEG and PEO, known for their low binding behavior) with different Mw values [36]. First, the PWP of the membranes was measured. Then, the feed solution was introduced and the filtrate was collected at 2 bar. During this step, the filter cell was stirred at 500 rpm to minimize concentration polarization. The membrane was then washed several times with DI water to remove the reversible solute adsorption and the PWP was measured again. The difference between the final and initial PWP is a measure of the irreversible fouling [17]. The recovery of the initial PWP meant that the membrane was not fouled during the retention experiments. The PEG and PEO contents in the feed solutions (*C_f_*) and permeate (*C_p_*) were determined by flow injection analysis using a 2414 refractometer (Waters Corporation). Solute rejection was calculated using the following equation:(3)$$R\left(\%\right)=\left(1-\frac{Cp}{Cf}\right)\times 100$$

## 3. Results and Discussion

### 3.1. Effect of Cross-Linking Reaction Time on Membrane Morphology

Membrane morphology is a very important feature as it influences many membrane properties. Figure 1 shows the evolution of the surface morphology of the composite membranes in the presence of the pore-forming agent PEG and after a 30-minure post-treatment with STPP. A uniform, smooth and flat surface with no detectable pores was observed in the case of the pristine CS membrane (CK0). In contrast, the composite membranes showed rougher surfaces with apparent KO particles. However, the penetration and intercalation of CS chains into the KO layers gives rise to a cohesive three-dimensional structure. A well-defined pore structure appears when the starting suspension formulation contains PEG, indicating that it plays the role of a pore-forming agent through its interaction with CS. Pore formation is then related to the release of PEG to the external solution during phase separation [21]. As can be seen in Figure 1, post-treatment with STPP led to a tightening of the surface pores. 

The effect of cross-linking on the top surface porosity and bulk morphology is further examined in Figure 2 (left), which compares the surface morphology of the untreated membrane (CK5-PEG) with that of the membranes that underwent a post-treatment with STPP for 15, 30 and 60 min. As can be seen, cross-linking significantly affected the top surface pore size and pore density, indicating that negatively charged STPP can easily diffuse into the interpenetrated positively charged CS chains. After immersing the CK5-PEG membrane in STPP solution, a decrease in pore size could be observed with reaction time, such that only a few pores could be detected in SEM micrographs after 60 min of reaction. It can also be noted that network reorganization requires some induction time. Indeed, after 15 min of contact, little change in surface pore size and density was observed with the starting membrane. In contrast, a noticeable reduction in surface pore size occurred after 30 min of immersion in the cross-linking bath (CK5-PEG-STPP30). These operating conditions allowed the preparation of composite membranes with circular, well-defined and much smaller surface pores. However, longer reaction times led to pore clogging and the appearance of some holes around the KO particles (CK5-PEG-STPP60). It can be concluded that the cross-linking makes the membrane surface evolve from a foam morphology to a dense layer. This result is in agreement with previous reports that have shown that long reaction times result in increased pore densification of a CS membrane due to the higher cross-linking density [37].

Figure 2 (right) shows micrographs of the corresponding membrane cross-sections. SEM observation revealed that the introduction of PEG into the KO/CS suspension resulted in a highly porous sponge-like structure with interconnected pores on the surface and the appearance of finger-like structures underneath. As mentioned above, similar structures characteristic of the non-solvent phase inversion method have been found for other types of composite membranes [6], and the mechanism of their formation has been explained by different authors [18,19]. Cross-linking resulted in membrane thickening without affecting the finger-like structure significantly, except in the case of the longer reaction time (CK5-PEG-STPP60). The gradual penetration of STPP and intercalation between CS chains through electrical interactions are assumed to lead to the expansion of the polymer network, resulting in thickening of the top layer and that of the membrane cross-section. Figure 3 shows the evolution of the average thickness of the composite membrane and that of the surface layer above the finger-like structure determined from the SEM images, as a function of contact time with the STPP solution.

A two-step process could explain these findings: first, a rapid diffusion of STPP inside the membrane resulting in a steep increase in its total thickness, followed by reorganization of the CS polymer chains, probably due to an exchange of the electrical interactions between ammonium groups with KO by those with STPP. This second step corresponds to the intensification of the cross-linking reaction and is associated with densification of the polymeric network and a steady increase in the thickness of the top layer.

Figure 4 shows the PSD determined by image analysis of SEM photos of the three membrane top surfaces prepared with respective cross-linking reaction times of 0, 15 and 30 min (Figure 2, left). Note that the image analysis of the membrane surfaces cross-linked for 60 min did not yield significant values due to the low number of measurable pores.

### 3.2. Pore Size Distribution (PSD)

As discussed in the previous section, the surface size decreased with increasing the cross-linking time. However, it can be observed in Figure 4 that the PSD remained similar for all samples. It only shifted as the cross-linking reaction increased. The peak maximum was taken as the characteristic pore size value for the prepared composite membranes. It varied from about 160 nm for the non-cross-linked CK5-PEG membrane to about 15 nm for a membrane cross-linked for 30 min (CK5-PEG-STPP30). As mentioned before, the pore size changed little during the first 15 min of the reaction (from 160 nm to about 125 nm for CK5-PEG-STPP15), in agreement with the assumption of an induction time mainly related to the movement of the polymer chains to allow the cross-linking reaction to occur. 

In conclusion, cross-linking by electric interactions between the anionic phosphate groups of STPP and the cationic protonated amino groups of CS led to dramatic morphological changes in the composite membranes—in particular, a densification of the top surface layer resulting in a significant reduction in pore size. These two parameters can have a significant influence on the surface physicochemical properties and the operating performances of the membranes (see the following sections). Thus, a variety of membranes with pore sizes ranging from microfiltration to ultrafiltration could be prepared by varying the cross-linking conditions by STTP. The discussion below focuses primarily on the properties of CK5-PEG-STTP30, which represents the composite membrane most impacted by cross-linking.

### 3.3. Surface Chemistry of the Composite Membranes

Figure 5 presents the elemental composition of the surfaces of the pristine CS membranes, the CK5 composite and the cross-linked CK5-PEG-STPP30 composite, determined by EDX microanalysis. The CS membrane composition showed mainly the presence of C, N and O elements, while additional signals corresponding to Si and Al elements were detected in the CK5 composite membrane. A new peak corresponding to phosphorus could be observed in the cross-linked membrane, confirming the incorporation of STPP in the composite membrane.

The ATR-FTIR spectra of STPP powder and the non-cross-linked and cross-linked composite membranes were compared to highlight their signature peaks and to investigate the chemical interactions taking place between the composite membrane components (Figure 6). One of the characteristic peaks of STPP powder found at 1212 cm^−1^ and attributed to the antisymmetric stretching vibrations of the PO^2-^ group could be observed in the spectrum of CK5-PEG-STPP30, while it was not present in that of CK5-PEG, indicating that the cross-linking agent was indeed introduced into the membranes. In addition, another band appeared in the spectrum of the cross-linked membrane at 1540 cm^−1^, attributed to the antisymmetric deformation N-H vibration of the NH_3_^+^ ion related to the protonated CS. Under the acidic conditions of cross-linking, strong electrical interactions between NH_3_^+^ cations and PO^2-^ anions can occur, causing a reduction in the intensity of the STPP peak at 1212 cm^−1^ compared to that of pristine STPP powder [35].

### 3.4. Thermal and Chemical Stability of the Composite Membranes

The thermal stability of the prepared composite membranes was determined by TGA. Figure 7 presents the typical thermograms obtained for the STPP powder and the non-cross-linked (CK5-PEG) and 30-minute cross-linked (CK5-PEG-STPP30) composite membranes. As can be seen, STPP can be considered thermally stable up to 800 °C. Three weight losses can be observed in the TGA curves of both CK5-PEG and CK5-PEG-STPP30. The first one between 50 and 150 °C is attributed to the release of water molecules more or less adsorbed within the composite membrane. The second weight loss between 200 and 400 °C corresponds to the partial decomposition of the CS and remaining PEG chains, while the third between 400 and 600 °C is assumed to result from the simultaneous dehydroxylation of kaolinite and final degradation of the CS chains [21]. Comparing the thermograms of these two samples, a lowering of about 50 °C can be observed for the decomposition temperature of the polymer moiety from 298 to 250 °C as a consequence of the cross-linking with STPP. A difference of the same order of magnitude can also be seen in the case of the inorganic part from 569 to 494 °C. The competition between the electrical interactions of CS with KO and those between CS and STPP required for the polymer cross-linking probably explains the observed decrease in thermal stability. Finally, the difference in residual weight at 800 °C of about 10% of the total weight between CK5-PEG and CK5-PEG-STPP30 reveals the amount of STPP included in the cross-linked membrane.

Resistance to water washout is an important parameter to consider when using these composite membranes in water treatment applications. Protonation of the amine groups in an acidic environment results in partial dissolution of CS, causing deterioration of the membrane material. Therefore, the water solubilization (WS) of the pristine CS membrane (CK0) and the non-cross-linked and cross-linked membranes under different pH conditions (pH = 2, 4 and 9 and at the natural pH of deionized water (6.2)) was investigated to examine their chemical stability in aqueous media (Table 1). As expected, the pure CS membrane had low chemical stability and dissolved at pH values below neutrality, while the composite membranes retained their original state at this pH. The association with KO prevented the dissolution of CS, confirming that the membrane did not correspond to a simple mixture but resulted from a strong interaction between the different components.

However, these interactions were weakened by decreasing the pH, which eventually led to the complete dissolution of the membranes at pH 4. Therefore, stabilization of the composite membranes at acidic pH is crucial for applications in aqueous media. The obtained data in Table 1 show that cross-linking with STPP can solve this issue. Indeed, the cross-linked composite membrane CK5-PEG-STPP30 was stable at acidic pH even when immersed in a medium with pH 2. The compact network formed by high numbers of inter-chain bonds between CS and STPP led to the formation of a three-dimensional network structure and the loss of chain mobility responsible for the improved chemical stability of the resulting membranes. Moreover, cross-linking greatly reduces the number of protonated amine groups available to induce CS dissolution.

### 3.5. Mechanical Properties of the Composite Membranes

Good mechanical properties are one of the requirements for practical applications of the membranes. Therefore, the tensile strength and elongation at break were measured in the wet state (Table 2). These properties are closely correlated with the membrane structure, as the results showed that the mechanical properties of the STPP-cross-linked composite membranes were affected by the cross-linking time. The tensile strength values increased significantly by lengthening the cross-linking time, with the CK5-PEG-STPP60 sample showing a value more than twice that of the non-cross-linked membrane. In a previous section, it was shown that the morphology of the composite membrane was profoundly altered upon cross-linking with STPP. The formation of the three-dimensional network between CS and STPP results in an increase in membrane thickness and densification of the structure. The increase in the mechanical strength of the cross-linked membranes can then be explained by the restriction of CS chain movements and the decrease in pore volume.

However, the opposite trend was observed for elongation at break. The elongation at break of the membranes decreased with increasing the cross-linking time, indicating an increase in the stiffness and brittleness of the membranes. This can again be explained by the increase in the number of inter-chain bonds as the reaction proceeded, reducing the flexibility of the network and, thus, the plasticity of the material. From these values, it is obvious that cross-linking plays a crucial role in improving the mechanical strength and reducing the elongation at break of the composite membranes. Despite these variations, the mechanical properties obtained remain well suited to the operating conditions of microfiltration and ultrafiltration.

### 3.6. Surface Properties of the Composite Membranes

The interfacial properties between the membrane and the feed solution strongly influence the performance of filtration in aqueous media. The hydrophilicity and surface charge can be determined by WCA and zeta potential measurements, respectively.

Table 2 shows the results of WCA measurements for the different prepared composite membranes, illustrating the effect of cross-linking on surface wettability. The obtained WCA values show a steady increase with treatment time, indicating that the surface became less and less hydrophilic. However, the difference with respect to the starting membrane remained small (ca 10°). This result is in agreement with the literature. The introduction of a CS cross-linking step has been reported to decrease the hydrophilic nature of the membrane [32]. This effect can be explained by the increase in the rigidity of the CS backbone and the reduction in the pore size, which prevent the access of water molecules to the CS polar group and also the decrease in the number of hydrophilic functions (amine and hydroxyl) available to yield solvation interactions with water.

The surface charge of CK5-PEG-STPP30 was determined in terms of zeta potential as a function of pH (Figure 8). The isoelectric point was found to be close to pH 7, meaning that the membrane surface was positively charged for pH values below 7 and negatively charged above. The protonation of the amine functions of CS (pKa about 6) is responsible for the positive value of the zeta potential in acidic media, whereas the negative charges of KO and STPP become predominant in basic media due to the deprotonation of the amine functions. It is noticeable that the zeta potential value (maximum about 15 mV) remained low over the whole pH range, confirming that the ionic functions were mainly involved in the interactions between the components of the composite membrane.

### 3.7. Permeation Properties of the Composite Membranes

Permeation properties, namely productivity and selectivity, were evaluated in terms of PWP and MWCO. It has been shown in previous work (Rekik et al., 2019) that the addition of PEG to a KO/CS suspension strongly improves the water permeability of the composite membranes obtained with this formulation owing to the sponge-like morphology of the surface layer and the presence of a finger-like through structure, as observed in Figure 2.

Figure 9 shows the effect of cross-linking time on *PWP* and *MWCO* values. As can be seen, the *PWP* values decreased with the reaction advancement. The aforementioned induction period of approximately 15 min led to only a slight decrease in water permeability despite the increased membrane thickness, confirming the previous findings. On the other hand, the reduction in permeability was more pronounced for longer reaction times. These results obtained under realistic membrane operating conditions are in agreement with the local determinations of morphological changes (pore size and top layer thickness) from SEM micrographs in Section 3.1 and Section 3.2. Clearly, the cross-linked composite membranes exhibited a more pronounced barrier to the passage of water molecules, resulting in a decrease in permeation flux. Interestingly, an excellent correlation was found between the PWP and the MWCO data determined for the corresponding membranes (Figure 9). It can be concluded that the dominant parameter accounting for the decrease in permeability is the progressive reduction in pore size by cross-linking with STPP. Other factors such as increased surface layer thickness leading to enhanced tortuosity may also contribute to the decrease in PWP [36]. Furthermore, the decrease in PWP as a function of cross-linking extent is also in good agreement with the increase in WCA values from CK5-PEG-STPP15 to CK5-PEG-STPP60.

The selectivity of the composite membranes in terms of MWCO confirms that the potential field of application can cover the range from microfiltration to ultrafiltration. Pore size tuning by STPP cross-linking could then allow the preparation of KO/CS composite membranes with specified properties in terms of productivity and selectivity by adjusting the reaction conditions.

## 4. Conclusions

Cross-linking of KO/CS composite membranes with STPP was successfully achieved in this work, which greatly improved their chemical stability in acidic media. The cross-linking density was found to drastically alter the morphology, mainly resulting in a decrease in the average pore size. The MWCO determination confirmed these observations. It was concluded that the filtration ability can be easily tuned from microfiltration (about 100 nm pore size after 15 min) to ultrafiltration (about 15 nm after 30 min) by adjusting the reaction time. The excellent mechanical properties, good pH stability and observed permeability offer a promising development for these sustainable and low-cost composite membranes in drinking water and wastewater treatment.

## Figures and Tables

**Figure 1 membranes-13-00229-f001:**
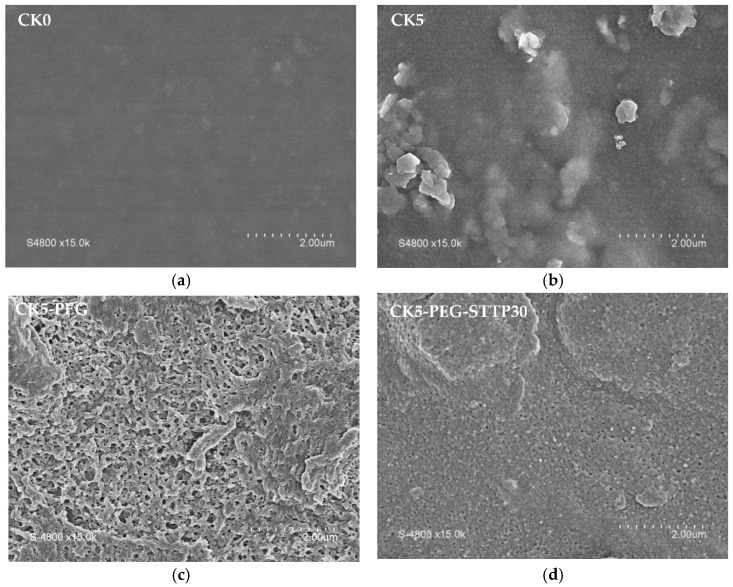
SEM pictures of the membrane surface (magnification × 15,000): (**a**) CK0, pristine chitosan membrane; (**b**) CK5, chitosan/kaolin composite membrane; (**c**) CK5-PEG, with PEG as a pore former; and (**d**) CK5-PEG-STPP30, cross-linked membranes (reaction time, 30 min).

**Figure 2 membranes-13-00229-f002:**
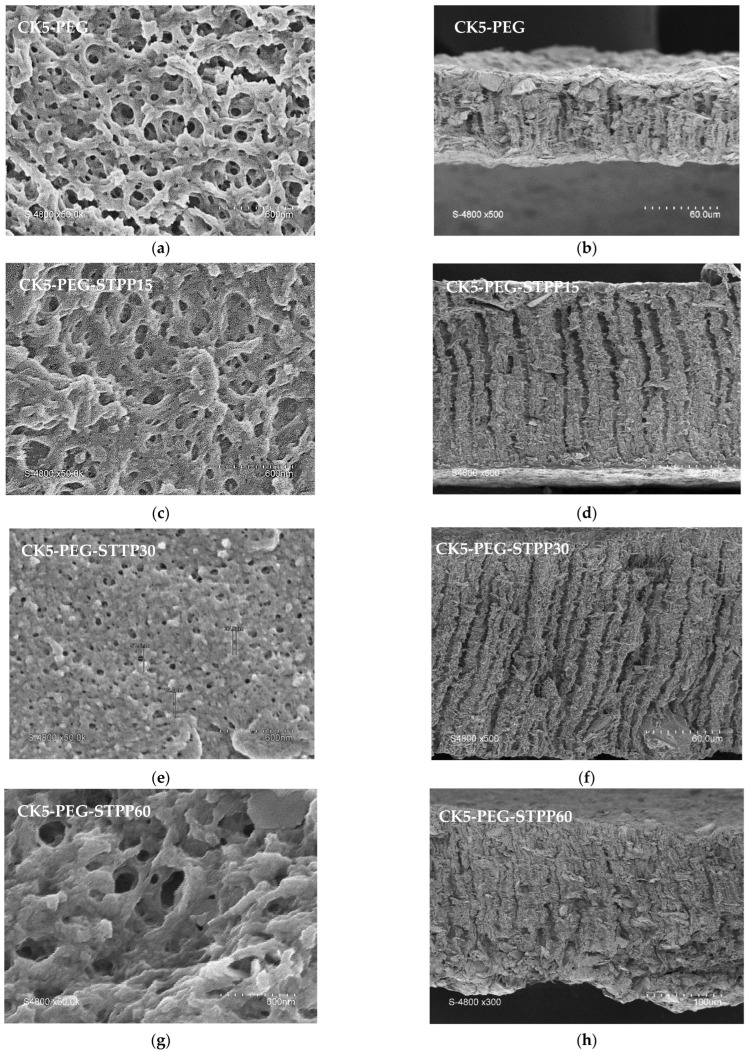
SEM pictures of the membrane surface after 0, 15, 30 and 60 min of reaction time with STPP (**a**), (**c**), (**e**), (**g**), respectively (**left**, magnification × 50,000) and membrane cross-section (**right**, magnification × 500 for (**b**), (**d**), (**f**) and × 300 for (**h**)).

**Figure 3 membranes-13-00229-f003:**
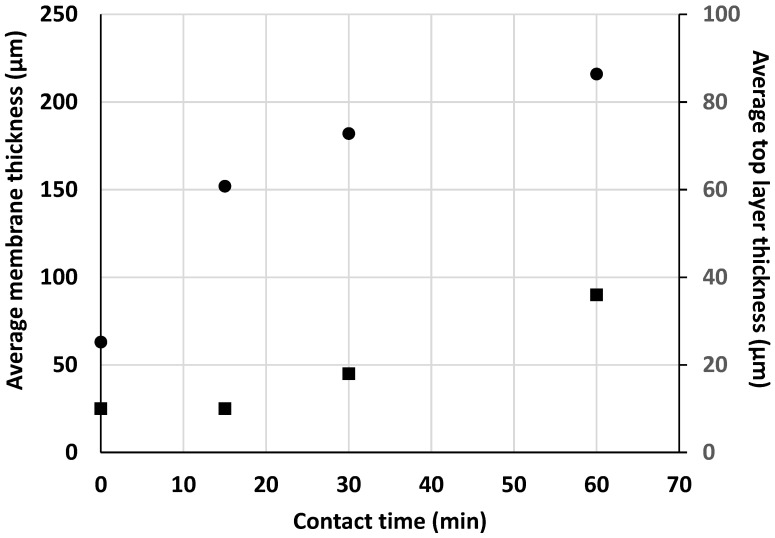
Variation of the average membrane (●) and top layer (■) thickness after 0, 15, 30 and 60 min reaction time with STPP.

**Figure 4 membranes-13-00229-f004:**
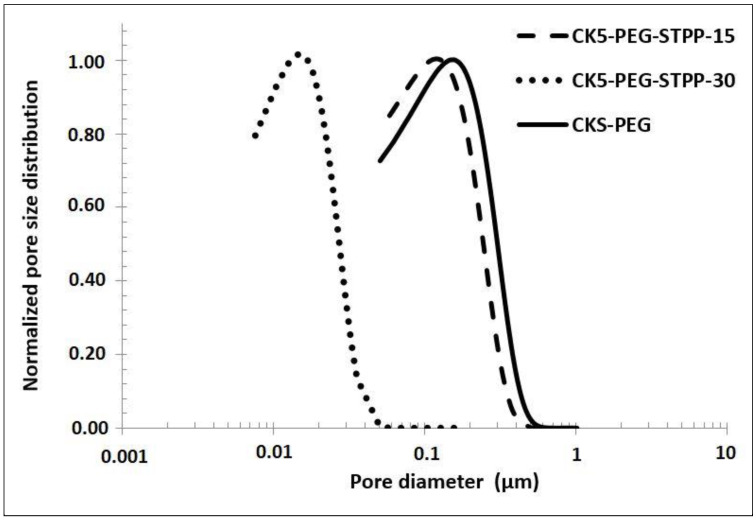
Pore size distribution determined by image analysis of SEM top surface of composite membranes after 0, 15 and 30 min reaction time with STPP.

**Figure 5 membranes-13-00229-f005:**
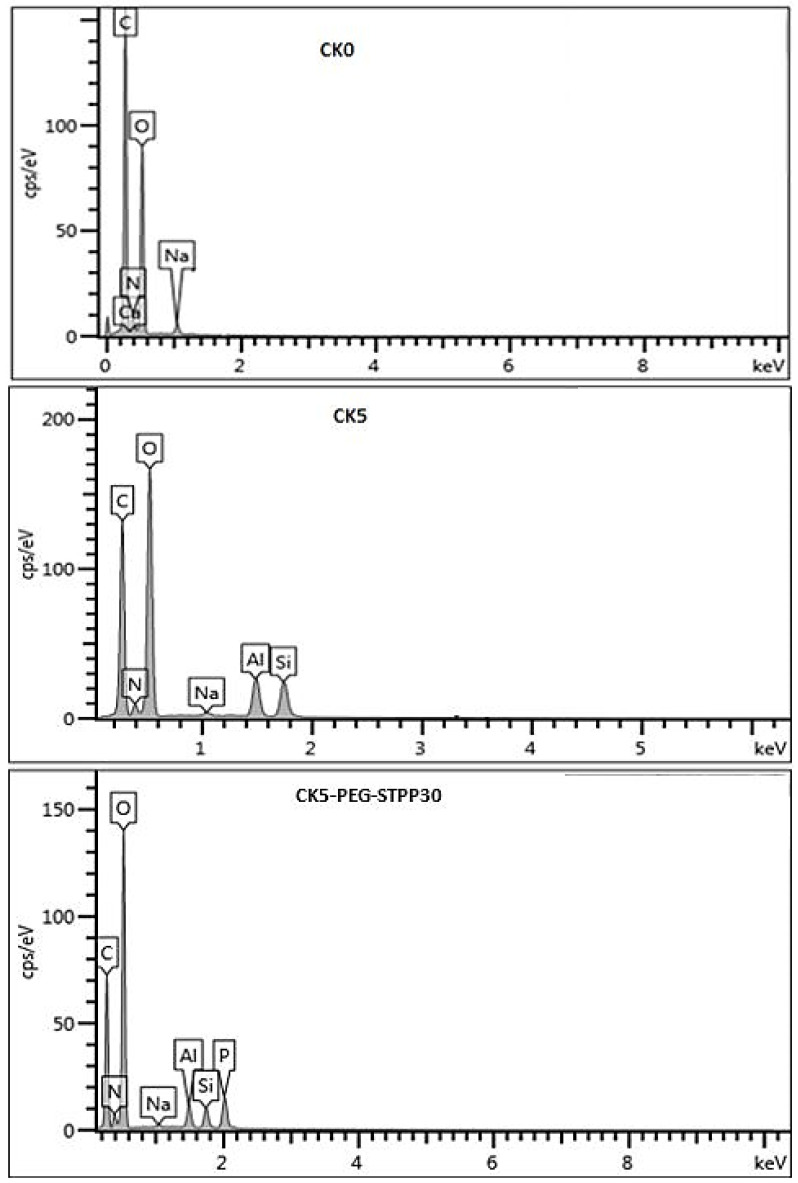
EDX microanalysis of pristine chitosan membrane (CK0), chitosan–kaolin composite membrane (CK5) and STPP cross-linked membrane (CK5-PEG-STPP30).

**Figure 6 membranes-13-00229-f006:**
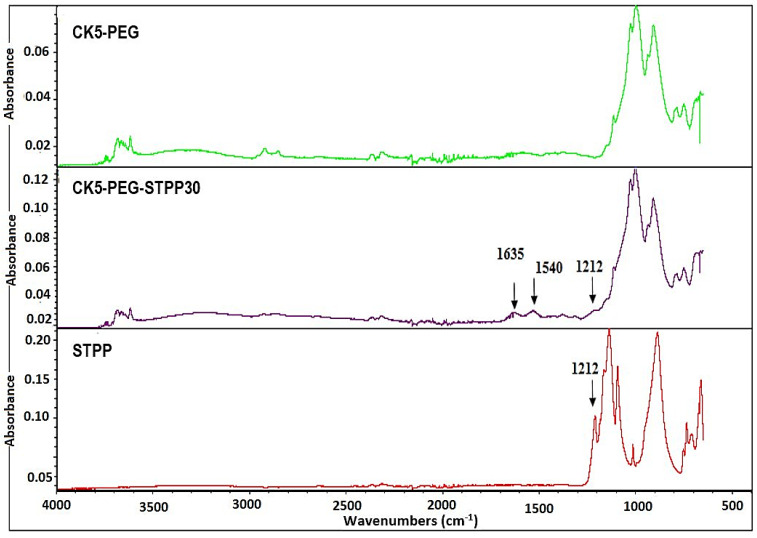
Comparison of ATR-FTIR spectra of CK5-PEG and CK5-PEG-STPP30 composite membranes and STPP powder.

**Figure 7 membranes-13-00229-f007:**
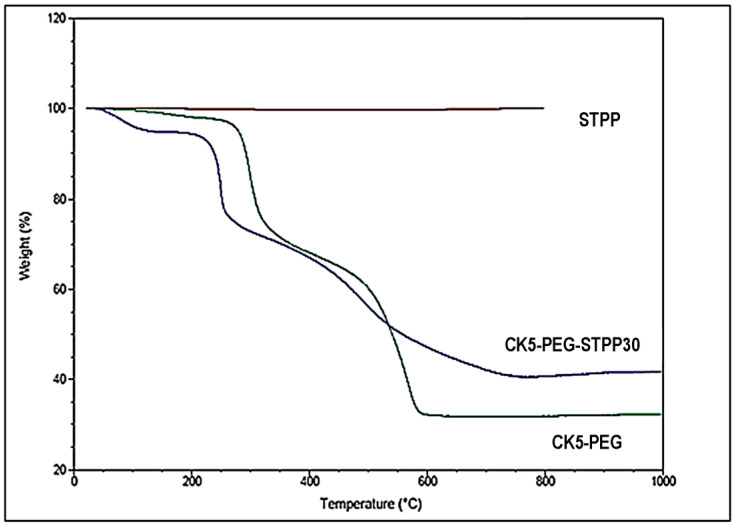
Thermograms of STPP powder, non-cross-linked CK5-PEG and 30-minute cross-linked CK5-PEG-STPP30 composite membranes.

**Figure 8 membranes-13-00229-f008:**
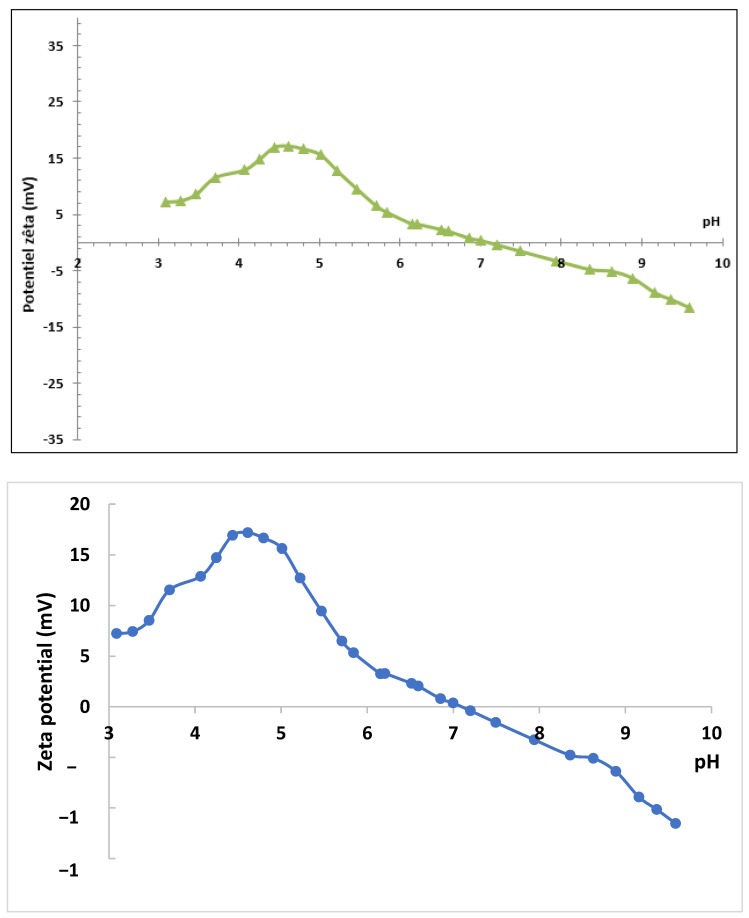
Zeta potential as a function of pH (
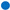
) for CK5-PEG-STPP30 composite membrane.

**Figure 9 membranes-13-00229-f009:**
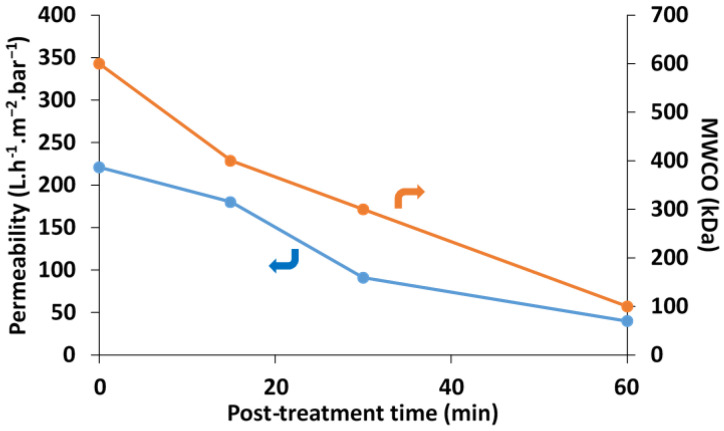
Variation in *PWP* and *MWCO* as a function of the cross-linking time.

**Table 1 membranes-13-00229-t001:** Solubility of KO/CS composite membranes compared to that of pristine CS membrane (CK0) in water at different pH.

	CK0	CK5-PEG	CK5-PEG-STPP30
pH = 9	Insoluble	Insoluble	Insoluble
DI water (pH = 6.2)	Soluble	Insoluble	Insoluble
pH = 4	Soluble	Soluble	Insoluble
pH = 2	Soluble	Soluble	Insoluble

**Table 2 membranes-13-00229-t002:** Mechanical properties and water contact angle of the obtained composite membranes.

Sample Name	Tensile Strength (MPa)	Tensile Strain at Break (%)	Water Contact Angle (°)
CK5-PEG	2 ± 0.03	8.2 ± 0.5	62.0 ± 0.3
CK5-PEG-STPP15	2.5 ± 0.1	6.0 ± 0.1	67.0 ± 0.1
CK5-PEG-STPP30	3.4 ± 0.3	3.8 ± 0.1	69.0 ± 0.3
CK5-PEG-STPP60	4.5 ± 0.4	2 ± 0.1	72.0 ± 0.4

## Data Availability

The data presented in this study can be requested from the corresponding author for a reasonable reason.

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
