# Peer review of "Performance Enhancement of Kaolin/Chitosan Composite-Based Membranes by Cross-Linking with Sodium Tripolyphosphate: Preparation and Characterization"

_membranes, 2023, doi:10.3390/membranes13020229_

Round 1

Reviewer 1 Report

The work entitled “Performance enhancement of Kaolin/Chitosan composite based two membranes by cross-linking with sodium tripolyphosphate: preparation and characterization” presents a study of obtaining Kaolin/Chitosan composite membranes using sodium tripolyphosphate as a cross-linking. The study presents meaningful information for the literature on membrane separation processes. I recommend its publication after some corrections. Still, I present some suggestions to the authors.

1- Some figures show information with low resolution, making it difficult to analyze (for example Figure 2).

2- Figure 9 can be reformulated in 2 Y axes, one with the PWP and the other with the MWCO.

3- The authors presented a pore size distribution based on SEM images. I suggest using techniques like mercury intrusion porosimetry, nitrogen adsorption techniques (BET, BJH...), and the Archimedes method (porosity).

Author Response

Reviewer #1: 

Indications and modifications in the text in response to the reviewer#1 are highlighted in yellow

  1. COMMENT:  Some figures show information with low resolution, making it difficult to analyze (for example Figure 2).

ANSWER:  In fact, the figures in the submitted manuscript had good resolution. The layout led to a loss of resolution. Modifications were done to reintroduce the original figures into the formatted document.

  1. COMMENT: Figure 9 can be reformulated in 2 Y axes, one with the PWP and the other with the MWCO.

ANSWER:  In the revised paper, Figure 9 was reformulated in 2 Y axes.

  1. COMMENT: The authors presented a pore size distribution based on SEM images. I suggest using techniques like mercury intrusion porosimetry, nitrogen adsorption techniques (BET, BJH...), and the Archimedes method (porosity).

ANSWER: We appreciate your suggestion regarding the different methods of determining pore size. However, the authors would like to make the following comments on the limitations of these techniques:

  • Mercury intrusion porometry is a well-suited technique for measuring pore diameter in mechanically stable membranes such as ceramic materials. For example, a pressure above 3.5 MPa is required for mercury to penetrate pores smaller than 0.5 µm, so polymer-containing membranes are generally not robust enough to withstand such pressures without collapsing (see table 2).
  • Nitrogen adsorption techniques are more suitable for the analysis of the mesoporous domain. In this case, capillary condensation at 77 K is generally accepted as responsible for pore filling. Two remarks are necessary. Again, the calculation requires that the pores be rigid and undeformable, which is generally not possible with polymer-containing membranes, because vacuum and heating steps are performed before the measurement. Second, the volume of nitrogen adsorbed by the top layer is very small, so it is generally impossible to determine the surface pore size of an asymmetric polymer-containing membrane.
  • For the same reasons as above, the pore size of the top surface, which is the most important parameter for membranes, cannot be obtained by the Archimedes method.

Therefore, image analysis of surface micrographs was used in this work to determine the pore size of the “top” membrane surface, as specified in the manuscript (the term “top” was added in several parts of the manuscript to clarify this point). Although the value determined by this method was obtained using several images from different membranes, it is obvious that it can only be considered as a local determination. The most significant parameter is the MWCO determined under the operating conditions. This limitation and the correspondence with the values determined for the MWCO was indicated in the text (p. 16, lines 489-491).

These results obtained under realistic membrane operating conditions are in agreement with the local determinations of morphological changes (pore size and top layer thickness) from SEM micrographs.

Reviewer 2 Report

In this work, the authors studied “Performance enhancement of Kaolin/Chitosan composite based membranes by cross-linking with sodium tripolyphosphate: preparation and characterization”. In this paper, the low-cost membranes based on readily available mineral and polymeric materials were developed from cast suspensions of kaolin and chitosan using aqueous phase separation and polyethylene glycol as a pore-forming agent. However, some of the explanations may need further illustration; and more importantly, there is scope to enhance the clarity of findings. I consider that the scientific discussion of this manuscript needs to be enhanced with proper justifications to be published in the Membranes Journal. Also, the authors should consider critically these comments to improve the quality of the work while revising the manuscript.

Comments:

  1. abstract needs to be revised as there is no output is mentioned.
  2. What advantages of the developed method have to be mentioned properly?
  3. The SEM images scale should be the same. Here 3 different scales are used.
  4. How authors controlled the fouling over the surface of the membrane?
  5. Authors can compare and cite these recent articles about fouling with clay-related work like Membranes 12 (2022) 768; J. Membr. Sci., 609 (2020) 118212 etc.
  6. There should be clear labelling and name on the ATR-FTIR spectra. Make a single overlayed graph.
  7. The conclusion is too gigantic write it the proper way.
  8. Polish the grammar once. There are several grammatical mistakes were observed.

Author Response

Reviewer #2: 

Indications and modifications in the text in response to the reviewer#1 are highlighted in green

  1. COMMENT: Abstract needs to be revised as there is no output is mentioned.

ANSWER: In the revised paper, the abstract (lines 13-29) was reorganized and reformulated to introduce the major outputs.

“A new family of environmentally friendly and low-cost membranes based on readily available mineral and polymeric materials has been developed from cast suspensions of kaolin and chitosan using aqueous phase separation and polyethylene glycol as a pore-forming agent. The as-fabricated membranes were further crosslinked with sodium tripolyphosphate (STPP) in order to strengthen the properties of the obtained samples. The functional groups determined by FTIR and EDX confirmed that the reaction occurred. A detailed study of the effects of crosslinking time on the physicochemical, surface and permeation properties showed that a 30 min reaction enabled the composite membrane to be stable in acidic media (up to pH 2) and increased the mechanical strength twofold compared to the non-crosslinked membrane. A similar morphology to that generally observed in polymeric membranes was obtained with a sponge-like surface overlaying a finger-like through structure. The top layer and cross-section thickness of membranes increased during STPP post-treatment while the pore size decreased from 160 to 15 nm. At the same time, the molecular weight cutoff and permeance decreased due to the increase in crosslinking density. These results observed in a series of kaolin/chitosan composite membranes showed that STPP reaction can provide control over the separation capability range, from microfiltration to ultrafiltration.”

  1. COMMENT: What advantages of the developed method have to be mentioned properly?

ANSWER: Introduction p.3, lines 117-136, was reorganized to properly mention the expected advantages of the developed post-treatment.

It is well known in the literature that a post-crosslinking treatment allows to increase the chemical and mechanical stability (Ref. cited in this paper for chitosan films and membranes:  Liu, Su, Lee, & Lai, 2005; Liu, Zhao, Pan, Bruggen, & Shen, 2016; Pozzo, da Conceição, Pinelli, Scharnagl, & Pires, 2018; Priyadarshi, Sauraj, & Singh Negi, 2018; Nowacki, Galiński, & Stepniak, 2019). The chemical stability in acidic media is indeed the main advantage pursued in this study as indicated in the introduction (p. 3, lines 117-118) in order to widen the field of applications of the obtained composite membranes.

The objective of this report is to investigate the crosslinking ability of CS in composite membranes to improve their chemical stability mainly in acidic media. The protonation of amine groups leads to partial dissolution of CS in water. It has been shown in our previous work that ionic interactions with KO shift the dissolution to a lower pH value but not enough to allow the use of such membranes in aqueous applications (Bouzid Rekik et al., 2017; 2019).”

Moreover, this reaction is also known to lead to a narrowing of the pore size (Ref. cited in this paper: Gassara et al., 2013; Wu et al., 2016). This last advantage allows to obtain a range of membranes with variable cut-offs by playing on the time of the cross-linking reaction.

 Another point to mention is that the crosslinking density leads to a tightening of the pore structure (Gassara et al., 2013; Wu et al., 2016). Therefore, this work aims at investigating whether by playing on the crosslinking time, it is possible to control the pore size and thus obtain a range of CS/KO composite membranes of variable selectivity.”

  1. COMMENT: The SEM images scale should be the same. Here 3 different scales are used.

ANSWER: We agree with the reviewer. The same magnification (x 50k) for SEM images of the membrane surfaces was used in Figure 2a,c,e,g and a magnification (x 500) for those of the membrane cross sections in Figure 2b,d,f.  Figure 3 shows that the thickness of the membrane increases with the cross-linking time, so that the entire cross-section of the membrane obtained after 60 minutes cannot be visualized at this magnification. Therefore, a different magnification (x 300) was used in Figure 2h and indicated in the caption.

  1. COMMENT: How authors controlled the fouling over the surface of the membrane?

ANSWER:  The fouling is indeed an important parameter to manage during MWCO determination. As indicated in the section “2.4.2. Molecular weight cut-off (MWCO) determination” (p. 6, lines 249-262), the used procedure is as follows:

“First, the PWP of the membranes was measured. Then, the feed solution was introduced and the filtrate was collected at 2 bar. During this step, the filter cell was stirred at 500 rpm to minimize concentration polarization. The membrane was then washed several times with DI water and the PWP was measured again. The recovery of the initial PWP means that the membrane was not fouled during the retention experiments.”

  1. COMMENT: Authors can compare and cite these recent articles about fouling with clay-related work like Membranes 12 (2022) 768; J. Membr. Sci., 609 (2020) 118212 etc.

ANSWER: We agree with the reviewer that the increase of surface hydrophilicity leading to resistance to fouling is also an important motivation for incorporating mineral particles in a hydrophobic synthetic polymer. This point was added (with the following reference) in the introduction (p. 2, line 66).

“For example, incorporating ceramic particles into a polymer matrix generally improves the membrane hydrophilicity, resulting in higher permeation flux and fouling resistance (Moucham, B.G. et al., 2022),”

Moucham, B.G., Sachin, K., Diksha, Y., Archana, Y., Neha, R.T., Swapnali, H., Hyung, K.L., & Pravin, G.I. (2022). Development of Antifouling Thin-Film Composite/ Nanocomposite Membranes for Removal of Phosphate and Malachite Green Dye. Membranes, 12(8), 768.

This work focused on the preparation and characterization of CS/KO composite membranes crosslinked with STPP. PEG molecules, known as low binding molecules, were used to determine the effective membrane MWCO.  Therefore, it was only verified that no irreversible fouling occurred to make sure that the experiments were properly conducted. Clarifications were inserted in the section “2.4.2. Molecular weight cut-off (MWCO) determination” (p. 6, lines 253 and 258).

“solutes (PEG and PEO known for their low binding behavior) with different Mw ... The difference between final and initial PWP is a measure of the irreversible fouling (Moucham, B.G. et al., 2022).”

  1. COMMENT: There should be clear labelling and name on the ATR-FTIR spectra. Make a single overlayed graph.

ANSWER: Changes were done as suggested.

  1. COMMENT: The conclusion is too gigantic write it the proper way.

ANSWER: In the revised paper, the conclusion (p. 17, lines 510-519) was shortened to present the key concerns raised.

“Crosslinking of KO/CS composite membranes with STPP was successfully performed in this work, which greatly improved their chemical stability in acidic media. The cross-linking density was found to drastically alter the morphology, mainly resulting in a decrease in the average pore size. MWCO determination confirmed these observations. It was concluded that the filtration ability can be easily tuned from microfiltration (about 100 nm pore size after 15 min) to ultrafiltration (about 15 nm after 30 min) by adjusting the reaction time. The excellent mechanical properties, good pH stability and observed permeability offer a promising development for these sustainable and low-cost composite membranes in drinking water and wastewater treatment.”

  1. COMMENT: Polish the grammar once. There are several grammatical mistakes were observed.

ANSWER: A native-english speaking colleague revised the manuscript.

Round 2

Reviewer 2 Report

It can be accepted now.